# Identification and In Vitro Evaluation of Milkfish (*Chanos chanos*) Frame Proteins and Hydrolysates with DPP-IV Inhibitory and Antioxidant Activities

**DOI:** 10.3390/foods14203456

**Published:** 2025-10-10

**Authors:** Anastacio T. Cagabhion, Wen-Ling Ko, Ting-Jui Chuang, Rotimi E. Aluko, Yu-Wei Chang

**Affiliations:** 1Department of Food Science, National Taiwan Ocean University, Keelung 20224, Taiwan; anastacioiii.cagabhion@wpu.edu.ph (A.T.C.III); sunnyko0414@gmail.com (W.-L.K.); m1321002@gm.ncue.edu.tw (T.-J.C.); 2Department of Home Economics, Western Philippines University, San Juan, Aborlan 5302, Philippines; 3Department of Food and Human Nutritional Sciences, University of Manitoba, Winnipeg, MB R3T2N2, Canada

**Keywords:** milkfish by-products, bioactivity, enzymatic hydrolysis, type 2 diabetes, antioxidant, DPP-IV inhibitory

## Abstract

The study presents the potential of milkfish frame, a by-product of milkfish processing, as a source of dipeptidyl peptidase IV (DPP-IV) inhibitory and antioxidant peptides with potential applications in type 2 diabetes management. Proteomic analysis identified key proteins, including 65 kDa warm temperature acclimation protein 1 and myosin heavy chain. In silico prediction (BIOPEP-UWM) guided the selection of proteases for generating DPP-IV inhibitory peptides. Enzymatic hydrolysates were produced and evaluated for bioactivity. Among the treatments, pepsin hydrolysis (2% *v*/*v*, 8 h) yielded the highest peptide content (283.64 mg/g), soluble protein (86.46%), and DPP-IV inhibitory activity (68.47%). The resulting milkfish frame pepsin hydrolysate (MFH) was further enhanced through ultrafiltration and simulated gastrointestinal digestion, which improved the DPP-IV inhibitory and antioxidant capacities. Cytotoxicity assays confirmed that MFH (0–100 μg/mL) was non-toxic to FL83B hepatocytes after 24 h. Moreover, treating TNF-α-induced FL83B cells with 10 μg/mL MFHs improved cell viability, reducing the toxicity induced by TNF-α in cells. These findings show that MFHs exhibit promising antidiabetic potential and could serve as natural alternatives to synthetic drugs for type 2 diabetes management. This also demonstrates the valorization of fish processing by-products into functional food ingredients, advancing sustainable approaches in food innovation.

## 1. Introduction

Diabetes is one of the most significant public health issues of the 21st century, affecting an estimated 537 million individuals worldwide. It is also estimated that this number will rise to about 783 million by 2045, with around 3.8 million deaths annually due to diabetes and related complications [1]. Over 95% of diabetic patients have type 2 diabetes, which is characterized by dysfunction in insulin secretion from pancreatic β-cells and insulin resistance. Early-stage diabetes can be managed through exercise and diet, while later stages require hypoglycemic drugs.

Food consumption stimulates the secretion of incretin hormones from the gastrointestinal tract in healthy individuals. This process promotes the synthesis and release of insulin while simultaneously inhibiting the production and release of glucagon, thereby helping to regulate blood plasma glucose levels. Conversely, patients with type 2 diabetes experience a reduction in incretin activity due to the enzymatic action of DPP-IV. The use of DPP-IV inhibitors mitigates this effect, thereby increasing incretin levels to enhance the regulation of glucose homeostasis. DPP-IV inhibitors have been demonstrated to improve glycemic control while ensuring an acceptable safety profile [2]. Clinical research has also shown that DPP-IV inhibitors lower HbA1c levels in individuals diagnosed with type 2 diabetes, where approximately 40% of participants achieved the HbA1c goal of less than 7%, with no greater hypoglycemia [3]. The administration of DPP-IV inhibitors in diabetic animal models has also been shown to promote the survival of beta cells, support the process of islet neogenesis, and increase insulin biosynthesis [4]. Moreover, in individuals with type 2 diabetes, DPP-IV inhibitors have been shown to enhance beta-cell function during fasting and postprandial states. These positive effects have been maintained in studies lasting up to two years [5]. In addition to its primary function in regulating incretin, DPP-IV exhibits a range of tasks across various organs. In the liver, DPP-IV contributes to the fibronectin-mediated interactions between hepatocytes and the extracellular matrix (ECM) [6].

The Food and Drug Administration (FDA) has approved sitagliptin, saxagliptin, linagliptin, and alogliptin as DPP-IV inhibitory drugs [7]. While synthetic DPP-IV inhibitors have demonstrated efficacy in diabetes management, they may also be associated with notable side effects, including headache and gastrointestinal symptoms such as nausea and diarrhea [8]. These considerations have encouraged increasing interest in identifying natural compounds with DPP-IV inhibitory potential from food-derived peptides. Such peptides, typically produced through enzymatic hydrolysis of dietary proteins, may exhibit DPP-IV inhibitory activity and possess additional physiological benefits such as antioxidant properties that help preserve β-cell function [2]. This dual bioactivity highlights the potential of food-derived peptides as functional food or nutraceutical ingredients that could complement conventional diabetes management approaches.

Milkfish (*Chanos chanos)* is a tropical marine fish primarily found in subtropical and tropical regions such as the Pacific Ocean, Indian Ocean, and Red Sea [9]. It is consumed by many as it is relatively cheap and nutritious [10]. It contains essential amino acids, unsaturated fatty acids, minerals, and water-soluble vitamins. Previous studies have reported that milkfish and its components exhibit potential antioxidant and metabolic-regulating activities [11]. Moreover, in 2023, the annual milkfish production in Taiwan is around 165,000 metric tons, with an average annual production value of about NT$100 billion (US$3 billion) [12]. Despite the substantial production volume, approximately 49.22% to 57.92% of this output consists of fish by-products [13] that remain underutilized, with an estimated 10% to 15% originating from bones and frames.

Recent studies have indicated that food processing by-products, including fish by-products, can serve as sources of bioactive peptides for the development of functional foods. Although several studies have reported bioactivities from various fish-derived proteins and peptides [14,15,16], studies on the bioactive potential of peptides obtained from milkfish frame (MF), a by-product commonly discarded during fish processing, are limited. Additionally, enzymatic hydrolysis of fish protein can produce peptides that exhibit a range of physiological activities, including antioxidant effects, blood sugar and blood pressure regulation, and antibacterial properties [17]. However, these bioactivities are species- and context-specific, underscoring the need for targeted studies. Protein hydrolysates hold potential for food applications as a natural promoting ingredient. While certain bioactivities have been reported from other milkfish sources [18,19,20], no specific studies have reported on the bioactivities of peptides from the milkfish frame (MF). Moreover, understanding how these bioactivities change during simulated gastrointestinal digestion is critical for assessing such peptides’ stability and functional potential. Therefore, this study aims to identify the MF proteins and bioactive peptides, to predict and analyze their bioactivities, primarily focusing on DPP-IV inhibitory and antioxidant activities, to evaluate the effects of the enzymatic hydrolysis, and simulated gastrointestinal digestion on these properties. The study also examines the protective potential of MFHs against TNF-α-induced oxidative stress in FL83B liver cells as a model for assessing their functional properties in food-related applications. It is hypothesized that peptide motifs predicted from MF protein isolates contain bioactive sequences that generate peptides with measurable DPP-IV inhibitory and antioxidant activities after enzymatic hydrolysis and gastrointestinal digestion. Through these objectives, the study contributes to the valorization of agri-food waste by extracting high-value compounds, promoting sustainability, and improving resource efficiency in food processing systems.

## 2. Materials and Methods

### 2.1. Materials

Milkfish frames (MF) were obtained from Hongyang Frozen Aquatic Products Co., Ltd., Chiayi County, Taiwan. Bromelain (EC 3.4.22.32), papain (EC 3.4.22.2), pepsin (EC 3.4.23.1), Gly-Gly-Gly, 1,1-diphenyl-2-picrylhydrazyl (DPPH), dipeptidyl peptidase IV (DPP-IV), dipotin A, F-12 Ham media were purchased from Sigma-Aldrich Co., St. Louis, MO, USA. FL83B was purchased from the Bioresource Collection and Research Center (RCBC), Hsinchu, Taiwan, and stored in liquid nitrogen. Tumor necrosis factor-alpha (TNF-α) was purchased from R&D Systems, Inc., Minneapolis, MN, USA. Cell Counting Kit-8 and Cytotoxicity LDH Assay Kit-WST were purchased from Dojindo Molecular Technologies, Inc., Rockville, MD, USA. All chemicals used in this study were of analytical grade.

### 2.2. Preparation of Samples

MF were thoroughly washed with distilled water and freeze-dried at −40 °C using an FD4.5 freeze dryer (INGMECH, Taipei, Taiwan) for 48 h. MF was homogenized with a high-speed pulverizer and stored at −20 °C. The freeze-dried MF powder underwent a degreasing process utilizing n-hexane at a ratio of 1:20 (*w*/*v*), with this extraction procedure being performed twice. The suspension was then filtered to remove the organic solvent, followed by a freeze-drying process at −40 °C for 48 h. It was then stored in a refrigerator at −20 °C for future applications.

### 2.3. Proximate Analysis

The prepared MF’s proximate composition was determined using the AOAC methods [21]. Moisture content was determined using oven drying at 105 °C (AOAC 950.46), crude protein was assessed using the micro-Kjeldahl procedure (AOAC 928.02), crude fat was analyzed using the Soxhlet extraction (AOAC 991.36), and crude ash was determined using the incineration in a muffle furnace at 550 °C (AOAC 920.153). Carbohydrate content was calculated by difference.

### 2.4. Preparation of Milkfish Frame Protein Isolate

Freeze-dried fish frame powder was combined with water in a 1:20 (*w*/*v*) ratio, and the pH was adjusted to 12 using 0.1 N NaOH [22]. The resulting mixture was subjected to electromagnetic heating and stirring for two h. The mixture was then centrifuged at 8000 rpm for 10 min using a high-speed centrifuge (Kubota, RA-2024, Bunkyo, Tokyo, Japan), and the supernatant was collected. The pH of the supernatant was then adjusted to 5.5 with 0.1 N HCl to facilitate the precipitation of myofibrillar protein [22]. The mixture was centrifuged again at 8000 rpm for 10 min, and the supernatant was collected and kept at −20 °C. The protein content was quantified using the Lowry method [23].

### 2.5. Sodium Dodecyl Sulfate-Polyacrylamide Gel Electrophoresis (SDS-PAGE)

The analysis was conducted according to the methodology established by the reference [24], utilizing a 12% separating gel and a 4% stacking gel. 0.1 g of the MF protein isolate was diluted in 1 mL of sample buffer, which comprised 3.55 mL of distilled deionized water, 1.25 mL of 0.5 M tris-HCl at pH 6.8, 2.5 mL of glycerol, 2 mL of 10% sodium dodecyl sulfate (SDS), 0.2 mL of 0.5% bromophenol blue, and 0.05 mL of β-mercaptoethanol. The mixture was heated in a dry bath at 95 °C for 5 min and centrifuged at 5000 rpm for 5 min. The supernatant collected was used as the sample solution for analysis. The electrophoresis chamber was filled with a 10× Running Buffer, composed of 3.03 g of Tris base, 14.4 g of glycine, and 10.0 g of SDS. A total of 5 μL of protein ladder and 10 μL of the sample analysis solution were loaded into the gel wells. The power supply was initially set to 70 V for the stacking gel and subsequently increased to 120 V for the resolving gel. Following electrophoresis, the gel was immersed in a staining solution of 1.5 g of Coomassie Brilliant Blue R-250, 500 mL of methanol, 100 mL of acetic acid, and 500 mL of distilled deionized water. The mixture was then gently agitated for 30 min. A destaining solution, composed of H_2_O, methanol, and acetic acid in a ratio of 7:2:1 (*v*/*v*/*v*), was employed to remove excess stain from the gel until the background became transparent.

### 2.6. Proteomic Approach and Mascot Database Comparison

#### 2.6.1. In-Gel Digestion

The protein bands of the SDS-PAGE gel with intense bands were taken using 200 μL pipette tips for in-gel trypsin digestion. Experimental protocols were modified from the method reported [24] with the following details: the gel pieces were placed into microcentrifuge tubes and destained twice with 80 μL 25 mM NH_4_HCO_3_/50% acetonitrile (*v*/*v*). To break the disulfide bonds in the proteins, the gel pieces were incubated with 15 μL of 20 mM dithiothreitol in 25 mM NH_4_HCO_3_ (pH 8.0) at 55 °C for 30 min and spun down at 5000 g speed at 10 °C for 3 min (3500, Kubota, Tokyo, Japan). After removing the supernatant, the gel pieces were alkylated with 15 μL of 55 mM iodoacetamide in 25 mM NH_4_HCO_3_ and incubated in the dark for 60 min at room temperature (26–28 °C). The excess iodoacetamide solution was removed by pipetting after the alkylation was finished. The destaining steps previously described were repeated twice with 100 μL of 25 mM NH_4_HCO_3_/50% acetonitrile (*v*/*v*). Then the gel pieces were dried using a solvent evaporator (SpeedVac Concentrator, Thermo Fisher, Asheville, NC, USA). The trypsin (modified porcine trypsin, sequencing grade, Promega, Madison, WI, USA) digestion was preceded by adding 15 μL of trypsin solution and 10 μL of 25 mM NH_4_HCO_3_ into each sample and incubating at 37 °C for six h without any chemical or physical means to terminate the reaction. After incubation, the tryptic peptides were extracted three times with 50 μL of extraction solution (50% acetonitrile/45% water/5% formic acid, *v*/*v*/*v*) for each extraction, and the extraction solution was combined and transferred into microcentrifuge tubes. The combined extraction solution was vacuum-dried using a solvent evaporator at room temperature (26–28 °C) and stored at −20 °C until liquid chromatography tandem mass spectrometry (LC-MS/MS) analysis.

#### 2.6.2. Liquid Chromatography-Tandem Mass Spectrometry (LC-MS/MS) Analysis

LC-MS/MS analysis for peptide identification was performed following the method described by the reference [25], with minor modifications. A volume of 40 μL of the mobile phase, comprising deionized water with 1% formic acid and acetonitrile in a ratio of 98:2, was introduced into a microcentrifuge tube to reconstitute the extracted peptide sample. The resulting mixture underwent ultrasonic agitation for 1 min and centrifugation at 15,000× *g* for 15 min. Subsequently, the supernatant was carefully pipetted and transferred into borosilicate glass vials. The obtained supernatant was subjected to analysis using ultra-high performance liquid chromatography with quadrupole time-of-flight mass spectrometry (UPLC/Q-TOF-MS/MS). The analytical column employed was a C18 BEH column (130 Å, 1.7 µm, 3 mm × 150 mm, Waters, Milford, MA, USA) operated at a 10 µL/min flow rate. This flow rate was selected to enhance peptide resolution and improve ionization efficiency for subsequent mass spectrometry analysis. The mobile phase consisted of double-distilled water (Buffer A) and acetonitrile (Buffer B). A gradient elution was executed for 50 min, during which the concentration of Buffer B was incrementally increased from 2% to 95% (Buffer A: double-distilled water; Buffer B: acetonitrile). The ion source was configured to operate at 2.8 kV in positive ionization mode at a temperature of 90 °C. Then, mass spectrometric analysis was conducted within the *m*/*z* range of 350–2000, capturing ion signals corresponding to charge states of 2+ to 6+.

#### 2.6.3. Tandem MS Database Comparison and Peptide Identification

The raw data obtained via the liquid chromatography tandem mass spectrometry (LC-MS/MS) were converted into PKL files using ProteinLynx Global Server v 2.4. The resulting files were compared with the NCBI database using Mascot MSMS ion search (http://www.matrixscience.com/cgi/search_form.pl?FORMVER=2&SEARCH=MIS) (accessed on 17 June 2025) [26]. The Mascot search conditions were Carbamidomethyl (C) and Oxidation (M) (variable modification), ±0.1 Da peptide mass tolerance, ±0.1 Da fragment mass tolerance, trypsin as the benchmark, and the molecular weight of all peptides was a single isotopic molecule (monoisotopic mass). Protein score was represented by the relative protein comparison identification index (Matrix Science, London, UK).

The proteins identified from the major gel bands were subsequently used to guide enzyme selection in the in silico BIOPEP analysis. Enzyme selection in BIOPEP reflected the proteolytic specificity relevant to the experimentally observed protein sequences, ensuring that the proteomic data directly informed predictions of bioactive peptide release.

To ensure data transparency and reproducibility, the resulting LC–MS/MS data have been deposited in the Zenodo public repository (CERN, OpenAIRE), with the digital object identifier (https://doi.org/10.5281/zenodo.17264290).

#### 2.6.4. In Silico Analysis of Bioactive Peptides by BIOPEP-UWM Database Tools

The BIOPEP-UWM database (https://biochemia.uwm.edu.pl/biopep-uwm/) (accessed on 30 June 2025) was employed to predict potential active peptide types, their quantities, physiological activities, and their respective positions within protein sequences [22]. Multiple enzymes, including ficain, papain, pepsin, chymotrypsin C, proteinase K, thermolysin, cathepsin G, and bromelain, were simulated to examine the active peptides produced during protein digestion facilitated by these enzymes.

### 2.7. In Vitro Analysis

#### 2.7.1. Enzymatic Hydrolysis of Milkfish Frame Protein Isolate

Following the simulation of enzymatic hydrolysis utilizing the BIOPEP-UWM database, the most appropriate enzymes (pepsin, papain, and bromelain) were chosen. Then, MF protein extract was combined with distilled water in a 1:100 (*w*/*v*) ratio and subjected to homogenization for five min [27]. Subsequently, the mixture was heated, and the pH was adjusted to the optimal level for enzymatic hydrolysis using 0.1 N NaOH or 0.1 N HCl, with specific pH requirements for different enzymes: pepsin at pH 2 and 37 °C, papain at pH 7 and 55 °C, and bromelain at pH 8 and 50 °C. Once the solution reached the desired conditions, the enzyme was introduced, and hydrolysis was conducted for 12 h, with pH adjustments made every 30 min throughout the process. Following hydrolysis, the resulting hydrolysate was neutralized and subjected to a water bath at 95 °C for 15 min to inactivate the enzyme. Subsequently, the mixture was centrifuged at 9600 rpm at 4 °C for 20 min, and the hydrolysate was collected. Finally, the hydrolysate was freeze-dried at −20 °C.

#### 2.7.2. Degree of Hydrolysis

The degree of hydrolysis (DH) was assessed utilizing the o-phthalaldehyde (OPA) method [28]. A fresh OPA solution was prepared by combining 5 mL of 20% (*w*/*w*) sodium dodecyl sulfate (SDS), 50 mL of 100 mM sodium tetraborate, 80 mg of OPA dissolved in 2 mL of methanol, and 200 μL of β-mercaptoethanol, which was subsequently adjusted to a final volume of 100 mL. Throughout the hydrolysis process, 10 μL of the enzyme hydrolysate was extracted hourly, to which 200 μL of OPA reagent was added and incubated for 100 s at room temperature. Then, the hydrolysates were combined with 6 N HCl and stirred for 24 h at 100 °C for total acid analysis. The absorbance was measured at 340 nm using a spectrophotometer (SpectraMax ABS Plus, Molecular Devices, San Jose, CA, USA). The peptide content was quantified through a standard calibration curve based on Gly-Gly-Gly. The degree of hydrolysis (DH) was determined using the equation:
(1)$$\mathrm{DH}\left(\%\right)=\left[\frac{{\left(NH_{2}\right)}_{T_{x}}{-\left(NH_{2}\right)}_{T_{0}}}{{\left(NH_{2}\right)}_{\mathrm{Total}}-{\left(NH_{2}\right)}_{T_{0}}}\right]\times 100\%$$

(NH_2_)_Tx_: Peptide content (mg/mL) at x min

(NH_2_)_T0_: Peptide content (mg/mL) at 0 min

(NH_2_)_Total_: Peptide content after acid hydrolysis (mg/mL)

#### 2.7.3. Peptide Concentration Determination

The peptide concentration was determined following the methods described by the reference [29] with appropriate modifications. Initially, the enzymatic hydrolysate was combined with an equal volume of 10% trichloroacetic acid and incubated for 10 min. Subsequently, the mixture was centrifuged at 4000 rpm for 10 min. Following centrifugation, 1 mL of the supernatant was extracted and combined with 4 mL of biuret reagent. This mixture was then incubated in the dark for 30 min, and the absorbance was measured at 340 nm using a spectrophotometer (SpectraMax ABS Plus, Molecular Devices, San Jose, CA, USA). The quantification of peptide content was obtained by comparing the absorbance to the standard curve generated from Gly-Gly-Gly.

#### 2.7.4. Dipeptidyl Peptidase IV (DPP-IV) Inhibitory Activity Assay

The hydrolyzed protein extracts were dissolved in a 100 mM Tris-HCl buffer at pH 8.0, to which 25 μL of a 1.6 mM solution of Gly-Pro-p-nitroaniline was added [30]. The resulting mixture was allowed to react in a 96-well plate at 37 °C for 10 min. Subsequently, 50 μL of DPP-IV was introduced, achieving a final concentration of 0.0025 U/mL, and the mixture was permitted to react at 37 °C for an additional 60 min. The reaction was subsequently halted by adding 100 μL of a 1 M sodium acetate buffer solution at pH 4.0, after which the absorbance was measured at 405 nm. The DPP-IV inhibitory activity was computed using the equation:
(2)$$\mathrm{DPP}-\mathrm{IV}\;\mathrm{inhibition}\;(\%)=[1-\frac{A405\left(\mathrm{test}\;\mathrm{sample}\right)\;-\;A405\left(\mathrm{test}\;\mathrm{sample}\;\mathrm{blank}\right)}{A405\left(\mathrm{positive}\;\mathrm{control}\right)-\;A405\left(\mathrm{negative}\;\mathrm{control}\right)}]\times 100\%$$ where A405(test sample): The absorbance value of the sample after reaction; A405(test sample blank): use Tris-HCl buffer (100 Mm pH 8.0) instead of the absorbance of DPP-IV; A405(positive control): Use Tris − HCl buffer (100 Mm pH 8.0) to replace the absorbance of the sample; A405(negative control): Use Tris − HCl buffer (100 Mm pH 8.0) instead of the absorbance of DPP-IV.

#### 2.7.5. Antioxidant Activity Assay

Antioxidant activity was analyzed using the DPPH radical scavenging activity assay. One hundred μL of MFH with a solid concentration of 10 mg/mL was combined with 0.1 mM of 2,2-diphenyl-1-picrylhydrazyl (DPPH) in ethanol [31]. The resulting mixture was thoroughly homogenized and incubated in the dark at room temperature for 30 min. The absorbance was subsequently recorded at 517 nm using a spectrometer. Distilled water served as the control, while glutathione at an equivalent concentration was employed as a positive control. The DPPH radical scavenging activity was determined using the equation:
(3)$$\mathrm{DPPH}\;\mathrm{radical}\;\mathrm{scavenging}\;\mathrm{capacity}\left(\%\right)=\frac{\left({\mathrm{OD}}_{517}\;\mathrm{control}-{\mathrm{OD}}_{517}\;\mathrm{sample}\right)}{{\mathrm{OD}}_{517}\;\mathrm{control}}$$ where OD_517_ control: the absorbance value of the control group; OD_517_ sample: the absorbance value of the sample group

#### 2.7.6. Fractionation

Fractionation was done through ultrafiltration. Filter membranes with different molecular weight cut-offs (MWCO) of 10 kDa, 5 kDa, and 1 kDa were subjected to a cleaning process involving shaking in double-distilled water for 15 s, which was repeated twice [32]. This was followed by shaking in a 20% alcohol solution for 15 s, repeated twice, and concluding with a final rinse in double-distilled water, again shaken for 15 s and repeated twice. The cleaned filter membrane was then positioned within the Amicon stirring ultrafiltration device. The MFH (1 mg/mL) was separated based on molecular weight at a temperature of 4 °C with a stirring speed of 300 rpm. The resulting fractions of 5–10 kDa, 1–5 kDa, and <1 kDa were subsequently freeze-dried and kept at −20 °C for later use.

#### 2.7.7. Simulated Gastrointestinal Digestion

The lyophilized protein extracts were diluted to a concentration of 5% (*w*/*w*) in double-distilled water, and the pH was modified to 2.0 using 1 N HCl [33]. Pepsin was added at an enzyme-to-substrate ratio of 1:25 (*w*/*w*) and incubated at 37 °C for two h. After the incubation, the enzyme was inactivated by heating in a water bath at 95 °C for 15 min. The pH was subsequently adjusted to 7.0 with 1 N NaOH, and pancreatin was introduced at the same enzyme-to-substrate ratio of 1:25 (*w*/*w*), allowing the reaction to proceed at 37 °C for four h. The pancreatin was then inactivated by heating at 95 °C for 15 min in the water bath. The digested sample was centrifuged at 9600 rpm at 4 °C for 20 min, and the supernatant was collected for further assessment of antioxidant activity and DPP-IV inhibitory activity.

#### 2.7.8. Cell Assay

##### FL83B Cell Culture

The nutrient mixture F-12 Ham media was mixed with 10% fetal bovine serum (FBS), 1% antibiotics, and 1.5 g/L NaHCO_3_ and was heated to 37 °C. Then, 10 mL of the prepared media was added to a 75T flask with a bottom area of 75 cm^2^. FL83B cells were removed from liquid nitrogen and thawed at 37 °C using a water bath before placing them in the 75T flask containing the media. The flask was then incubated with 5% CO_2_ at 37 °C [34]. The cell growth was observed daily using a microscope (Olympus Co., Tokyo, Japan).

Once the cell confluence reached 70–80%, the media was aspirated from the 75T flask, and the cells were rinsed with phosphate-buffered saline (PBS). One mL of trypsin (0.25% Trypsin, 0.53 mM, EDTA) was added to digest the cells and was placed in the flask. It was then stored in an incubator at 37 °C with 5% CO_2_ for 3 min. After digestion, 9 mL of media was added to terminate the reaction. The cell-containing media were then transferred to a 50 mL centrifuge tube and subjected to centrifugation at 3000 rpm for 5 min. The supernatant was then aspirated, and the cell pellet was resuspended in 1 mL of media. A quarter of the cell suspension was taken and added back to the 75T flask, along with fresh media.

##### CCK8 Assay

A commercially available kit (Cell Counting Kit-8 Dojindo Molecular Technologies, Inc., Rockville, MD, USA) was used to measure the release of electron mediator reduced form in the media [35]. FL83B cells were digested and centrifuged to calculate the cell concentration. A 5 × 10^4^ cells/mL solution was prepared and was added with 100 μL per well to a 96-well plate for 24 h. After which, the supernatant was aspirated and MFHs were added at concentrations of 0, 10, 25, 50, 75, and 100 μg/mL, or TNF-α at concentrations of 0, 10, 20, 30, 40, 50, and 75 ng/mL to the 96-well plate and incubated for 24 h. Then, 10 μL of the reaction reagent was added to each well, and the absorbance was measured at 490 nm after two h. The control group, without MFHs, was set as 100% to determine whether MFHs are toxic to liver cells, and the concentration of TNF-α causing liver cell damage was assessed.

After confirming the TNF-α conditions and assessing the absence of toxicity from MFHs, MFHs were explored to determine if they can reduce liver damage in the context of insulin resistance. The cell concentration of FL83B cells was calculated after digestion and centrifugation. A suspension of 5 × 10^4^ cells/mL was prepared, and 100 μL was added to each well of a 96-well plate and adhered to the plate for 24 h. After 24 h, the supernatant was aspirated, and MFHs at concentrations of 0, 10, 25, 50, 75, and 100 μg/mL were added to the 96-well plate. After another 24 h incubation, the supernatant was aspirated, and 75 ng/mL TNF-α was added to the 96-well plate for a 24 h reaction. Subsequently, 10 μL of the reaction reagent was added to each well, and the absorbance was measured at 490 nm after 2 h. The control group, which did not receive MFHs, was set as 100%, and the data values of different hydrolysate concentrations were converted into relative percentages for comparison.

##### Lactate Dehydrogenase (LDH) Assay

A commercially available kit (Cytotoxicity LDH Assay Kit-WST, Dojindo Molecular Technologies, Inc., Rockville, MD, USA) was used to measure the release of lactate dehydrogenase (LDH) in the medium [36]. The FL83B cells were digested and centrifuged at 400× *g* at 4 °C for 4 min to calculate the cell concentration. A suspension of 5 × 10^4^ cells/mL was prepared, where 100 μL was added to a 96-well plate. After 24 h, the supernatant was aspirated, and MFHs or TNF-α were added at concentrations of 0, 10, 25, 50, 75, and 100 μg/mL, or 0, 10, 20, 30, 40, 50, and 75 ng/mL, respectively, to the 96-well plate and were incubated for 24 h. Then, 100 μL of the supernatant was aspirated from each well and transferred to a new 96-well plate along with 100 μL of the reaction reagent. The plate was kept in the dark at room temperature for 30 min. Then, 50 μL of the terminator was added, and the absorbance was measured at 450 nm. The MFH was then evaluated for its toxicity to the liver cells, and the conditions under which TNF-α induced hepatocellular injury were determined.

After confirming that the MFHs are nontoxic, it was evaluated whether they could reduce liver damage in insulin resistance. FL83B cells were digested and centrifuged at 400 g at 4 °C for 4 min to calculate cell concentration, where 5 × 10^4^ cells/mL, and 100 μL/well was added to a 96-well plate and adhered to the plates for 24 h. After 24 h, the supernatant was removed, and 0, 10, 25, 50, 75, 100 μg/mL MFHs were added to a 96-well plate and incubated for 24 h. After removing the supernatant, 75 ng/mL of TNF-α and 100 μL of reaction reagent were added to the wells. After 30 min, it was incubated in the dark at room temperature, and 50 μL of terminator was added. The absorbance was then read at 450 nm. The control group, without MFHs, was set as 100% and the data values of each group were converted into relative percentages for comparison.

### 2.8. Statistical Analysis

The experimental data were analyzed statistically using SPSS software version 12, and the differences between samples were assessed by analysis of variance (ANOVA) with Duncan’s multiple range test.

## 3. Results and Discussion

### 3.1. Composition of Milkfish Frame

Results showed that the milkfish frame (MF) yield after freeze-drying was 36.30%. As shown in Table 1, proximate composition obtained for the MF powder aligns closely with the values reported for similar samples [37], indicating that fish frames are rich in protein and fat. Furthermore, after degreasing, the product’s protein content has increased relatively. Protein-rich substrates, like MF, are considered promising sources for generating bioactive peptides with potential physiological functions [38].

### 3.2. Milkfish Frame Protein Extraction

After defatting the MF using hexane, pH was adjusted to the protein’s isoelectric point using an acid-base extraction method, leading to protein precipitation and the formation of MF protein isolate. The soluble protein, peptide content, and yield of the protein isolates are summarized in Table 2. The protein content of the isolate after extraction has increased by 2.10%, confirming that the isolation process effectively concentrated the protein fraction. The observed increase can be attributed to removing non-protein components such as lipids, minerals, and water-soluble impurities during defatting and washing. At the isoelectric point, the reduction in electrostatic repulsion among protein molecules facilitates their aggregation and subsequent precipitation, allowing soluble non-protein components to remain in the supernatant. Consequently, the resulting protein isolate exhibits improved purity and higher protein concentration, which enhances its potential as a substrate for enzymatic hydrolysis and bioactive peptide production.

### 3.3. Protein Identification

From the results of the SDS-PAGE, the molecular weights of the MF protein isolate were mainly distributed between 180 kDa and 35 kDa (Figure 1). The hydrolyzed products (lanes B, C, and D) exhibited bands below 35 kDa, indicating that MF hydrolysates were mainly composed of low-molecular-weight peptides. This reduction in molecular weight is attributed to enzymatic cleavage of peptide bonds, which breaks the protein into shorter peptide chains and free amino acids [39].

In-gel digestion was performed on protein bands A1, A2, and A3 (Figure 1, lane A) of the MF protein isolate. The resulting tryptic peptides were analyzed using LC-MS/MS analysis to determine their peptide sequences. During mass spectrometry analysis, precursor ions with the highest signal intensities are selected for secondary fragmentation. After the LC-MS/MS analysis, several peptides were identified. As one of the representative examples, the peptide with a mass-to-charge ratio of 622.34 *m*/*z* was selected for collision-induced dissociation (CID) and subsequently fragmented in the collision cell (Figure 2a). Fragment ions were assigned according to the conventional b- and y-series nomenclature, representing the N-terminus (b-series) and the C-terminus (y series). Through the analysis and computation of the intervals between the fragments, a peptide sequence of 622.34 *m*/*z* was identified, corresponding to the tryptic peptide sequence VHVDALTAHGDDVVYAFR (622.344500,3+).

The tryptic peptide sequence identified through mass spectrometry was analyzed and compared with the Mascot database (Figure 2b). The peptide sequence VHVDALTAHGDDVVYAFR corresponded to the 65 kDa warm temperature acclimation protein 1 (WAP65-1) position, where the 244–261 amino acids in its sequence are the same. The remaining, red-highlighted regions in Figure 2b represent additional tryptic peptides that matched the WAP65-1 sequence. The proportion of matched peptides relative to the entire protein sequence is called sequence coverage, representing the ratio of the tryptic peptide identified by mass spectrometry to the whole protein sequence. Generally, an increase in sequence coverage correlates with an enhanced likelihood of accurately identifying the protein. Similarly, tryptic peptides obtained from bands A1, A2, and A3 were identified as the myosin heavy chain, as shown in Figure 2b (Accession number: ABK59967.1).

Figure 3a demonstrates the potential bioactive peptides in the WAP65-1 sequence identified from the MF protein isolate, while Figure 3b shows the potential bioactive peptides in the myosin heavy chain sequence. WAP65-1 and myosin heavy chain contained peptides with predicted DPP-IV inhibitory and antioxidant activity, such as active peptides with VW, VY, MY, LY, AY, and IR sequences. This peptide consists of 18 amino acids and includes several hydrophilic (Asp, His) and hydrophobic residues (Val, Ala, Phe), suggesting balanced polarity, which may influence its solubility and potential bioactivity. Aromatic and branched-chain amino acids like Val, Tyr, and Phe are also often associated with antioxidant and other functional bioactivities [40], supporting the potential health-promoting effects of peptides derived from MF proteins.

### 3.4. In Silico Analysis by BIOPEP-UWM

BIOPEP was used to predict the number of bioactive peptides to be released from identified MF proteins. Table 3 presents the potential bioactive peptides derived from the myosin heavy chain and WAP65-1. The data indicate that both proteins are rich in DPP-IV inhibitory peptides, with antioxidant peptides also present. Furthermore, simulations of enzymatic hydrolysis conducted using various common enzymes via the BIOPEP-UWM database tool revealed that pepsin yielded the highest number of DPP-IV inhibitory peptides, specifically 15 for the myosin heavy chain and 46 for WAP65-1. Based on these results, pepsin was used for the subsequent hydrolysis. Moreover, papain was also used since it came in second in the predicted DPP-IV inhibitory peptides. This enzyme was also considered, as it can selectively hydrolyze peptide bonds, particularly those involving proline, a common feature in DPP-IV inhibitory peptides [41]. Additionally, bromelain was further used for analysis as this enzyme is readily available and relatively inexpensive compared to many other proteolytic enzymes, aside from having the DPP-IV inhibitory and antioxidant activities, thus having the potential for further industrial applications.

### 3.5. Peptide Content During Enzymatic Hydrolysis

The results showed that peptide content increased progressively with hydrolysis time, stabilizing after 8 h with no significant difference (Figure 4a). A previous study indicated that prolonged hydrolysis might lead to the re-hydrolysis of bioactive peptides, reducing their effectiveness [38]. Considering industrial scalability and process efficiency, 8 h was selected as the optimal hydrolysis time.

Furthermore, the optimal enzyme-to-substrate ratio was investigated by testing pepsin, papain, and bromelain at concentrations of 1.0%, 1.5%, 2.0%, and 2.5%. Pepsin exhibited the highest peptide content at 283.64 mg/g at a 2% enzyme concentration (Figure 4b). The soluble protein content and yield of hydrolysates were measured using the Lowry method (Table 4). Similarly, pepsin also yielded the highest soluble protein content (86.46 mg/g), followed by bromelain (61.28 mg/g) and papain (59.44 mg/g). These findings suggest that pepsin’s cleavage specificity toward aromatic and hydrophobic residues enhances the release of both peptides and soluble proteins from the substrate. The results indicate that enzymatic hydrolysis can effectively increase the myosin and soluble protein, with 8 h of hydrolysis at a 2% enzyme-substrate ratio being the optimal condition.

### 3.6. Hydrolysis Rate of Hydrolysate

The hydrolysis rate increased rapidly in the first two hours, likely due to the cleavage of a significant amount of protein into small molecular peptides during the initial phase (Figure 5). This rapid phase likely reflects the accessibility of readily cleavable peptide bonds on the protein surface. After 12 h, pepsin showed the highest hydrolysis rate at 50.56%, outperforming papain and bromelain. These results show the role of enzyme specificity in determining hydrolysis efficiency, as pepsin’s preference for cleaving peptide bonds involving hydrophobic amino acids may facilitate more extensive protein degradation. A higher hydrolysis rate indicates a greater enzymatic capacity to break down protein substrates into free amino acids or low-molecular-weight peptides, which may enhance the bioactive potential of the resulting hydrolysates [42].

### 3.7. DPP-IV Inhibitory Activity

The unhydrolyzed MFH exhibited only 24.32% DPP-IV inhibitory ability (Figure 6a). However, after hydrolysis with pepsin, the inhibitory ability increased significantly to 68.47%, which was markedly higher than the other two enzymes. These results indicate that the hydrolysis using the three enzymes can enhance the DPP-IV inhibitory ability, with pepsin hydrolysate showing the best performance. The positive control, diprotin A, demonstrated a DPP-IV inhibitory ability of 94.23%. This indicates a potent inhibitory activity due to its sequence Ile-Pro-Ile, which contains hydrophobic amino acids and proline [43]. It has been demonstrated that peptides containing alanine or proline have stronger DPP-IV inhibitory activity [44]. Pepsin hydrolysate also contains higher proline content, contributing to its superior DPP-IV inhibitory ability [45]. Based on these results, pepsin hydrolysate was selected for subsequent ultrafiltration and in vitro simulated gastrointestinal digestion studies.

Molecular weight was inversely correlated with the DPP-IV inhibitory activity of peptides, indicating that short, low-molecular-weight peptides play a key role in this bioactivity [46]. As presented in Figure 6b, the DPP-IV inhibitory activity of the hydrolysate after ultrafiltration using a 10 kDa molecular weight cut-off (MWCO) membrane was not significantly different from that of the hydrolysate without ultrafiltration, with the highest activity reaching 86.49%. Previous studies have reported that DPP-IV inhibitory peptides derived from food proteins generally consist of 2–8 amino acid residues and have molecular weights between 0.2 and 1 kDa [47], suggesting that fractions below 1 kDa exhibit better DPP-IV inhibitory ability [48]. Although the <1 kDa fraction from unhydrolyzed samples exhibited inherent DPP-IV inhibitory activity, enzymatic hydrolysis was employed to further enhance this activity by liberating latent bioactive peptides.

Following this, simulated gastrointestinal digestion (SGID) was conducted to assess peptides’ stability and functional integrity under simulated physiological conditions. This provides a comprehensive evaluation of peptide generation, activity, and stability, essential for their practical application. No significant difference in DPP-IV inhibitory ability was observed between the undigested hydrolysate and the hydrolysate after the first stage of digestion (Figure 6c). However, after 4 h of pancreatin hydrolysis, the DPP-IV inhibitory ability improved to 83.79%. These results suggest that during in vitro SGID, longer peptides can be further cleaved into smaller fragments, and the pepsin hydrolysate now contains a higher proportion of proline residues at peptide termini, enhancing the generation of more potent DPP-IV inhibitory peptides. Furthermore, the maintained or improved inhibitory activity after digestion indicates that the hydrolysate is relatively resistant to enzymatic degradation, demonstrating high stability under gastrointestinal conditions. This stability is essential for potential functional food or nutraceutical applications, as bioactive peptides must remain intact or retain activity to exert physiological effects after oral ingestion.

### 3.8. DPPH Free Radical Scavenging Ability

Enzymatic hydrolysis enhanced MFH’s DPPH free radical scavenging capacity, with papain hydrolysate exhibiting the highest activity at 23.04% (Figure 7a). These results align with previous literature, indicating that DPPH free radical scavenging ability is better after being hydrolyzed by the papain enzyme [49]. Furthermore, ultrafiltration using a 1 kDa MWCO membrane further increased the antioxidant capacity of the hydrolysates (Figure 7b). This observation aligns with literature indicating that fish-derived antioxidant peptides typically have molecular weights between 0.5 and 1.5 kDa, and smaller peptides generally exhibit better antioxidant activity [50].

Post-SGID analysis enhanced DPPH radical scavenging activity following digestion (Figure 7c). Specifically, papain hydrolysate increased from 23.05% to 30.27%, pepsin from 19.82% to 26.83%, and bromelain from 20.24% to 25.10%. This improvement is likely due to the enzymatic release of peptides containing aromatic and hydrophobic amino acids, which are known to enhance radical scavenging activity. The hydrolysis of hydrophobic -COOH and -NH_2_ amino acids such as Phe, Tyr, Trp, Leu, Ile, Val, and Met makes peptides more accessible to DPPH free radicals, facilitating the trapping of free radicals and thus enhancing antioxidant properties [51]. These results suggest that both enzyme selection and peptide molecular size are critical factors influencing the antioxidant properties of MFH.

### 3.9. Cell Viability and Protective Effects of MFHs Against TNF-α-Induced Cytotoxicity

In diabetes, chronic inflammation and oxidative stress are frequently associated with cellular damage and apoptosis, especially within insulin-sensitive tissues, including the liver [52]. This loss of cellular integrity impairs the tissue’s capacity to effectively manage glucose and lipid metabolism, contributing to insulin resistance and metabolic dysfunction. Therefore, maintaining high cell viability under inflammatory conditions is crucial for preserving metabolic homeostasis.

The results in Figure 8a show that FL83B liver cells treated with MFHs at concentrations between 0 and 100 μg/mL for 24 h exhibited no significant changes in cell viability, indicating that MFHs are non-toxic and well-tolerated by liver cells. Consistently, LDH leakage analysis revealed no significant increase in extracellular LDH activity following MFH treatment. These results show that MFHs demonstrated a protective effect against TNF-α-induced cytotoxicity in FL83B cells, indicating their potential to support cell survival and maintain hepatic function under inflammatory stress, vital for effective glucose uptake and lipid metabolism.

The survival rate of cells significantly decreased when induced by TNF-α, particularly at a concentration of 75 ng/mL, where the cell survival rate dropped to around 80%. (Figure 8b). In the toxicity test, it was observed that 75 ng/mL of TNF-α led to an increase in LDH release, with a toxicity value of 118.03%, significantly higher than that of the group without TNF-α. This indicates that a 75 ng/mL concentration of TNF-α causes notable cell damage. Therefore, 75 ng/mL of TNF-α was selected to induce FL83B cell damage, simulating the effects on liver cells under insulin resistance for subsequent experiments.

Results showed in the cell survival assay that the control group, which received neither MFHs nor TNF-α, exhibited 100% viability (Figure 8c). The group treated with TNF-α alone showed a reduced survival rate of 84.15%, indicating substantial cytotoxicity. Co-treatment with 10 μg/mL MFHs mitigated TNF-α-induced cell death, increasing cell survival to 94.27%. Additionally, results showed that the TNF-α-only group showed a toxicity value of 118.03%, significantly higher than that of the group treated with 10 μg/mL MFHs. These results demonstrate that MFHs at this concentration can protect FL83B cells against TNF-α-induced cytotoxicity, highlighting their potential cytoprotective effects under inflammatory stress conditions.

Although these results demonstrate promising protective effects of MFHs in FL83B hepatocytes, it is essential to acknowledge the limitations of this in vitro model. These experiments do not directly assess DPP-IV activity or incretin signaling pathways, such as GLP-1 secretion or downstream insulinotropic effects. Furthermore, the model cannot fully capture the complex interplay of metabolic signals contributing to insulin resistance, including lipids and glucose. Therefore, future studies using diabetic animal models or co-culture systems are warranted to validate the anti-diabetic potential of MFHs under conditions that more closely mimic in vivo metabolic stress and allow direct measurement of GLP-1 secretion and cellular DPP-IV activity.

## 4. Conclusions

The study shows the potential of MF as a valuable source of bioactive proteins and hydrolysates with notable DPP-IV inhibitory and antioxidant activities. LC-MS/MS analysis identified the peptide sequence VHVDALTAHGDDVVYAFR (622.344500,3+) corresponding to WAP65-1. Results during peptide prediction showed that myosin heavy chain and WAP65-1 mainly contain DPP-IV inhibitory and antioxidant activity. Pepsin hydrolysis using 2% pepsin for 8 h yielded peptides with the highest inhibitory capacity, further enhanced by ultrafiltration and simulated gastrointestinal digestion. Moreover, adding 10 μg/mL of milkfish frame hydrolysates (MFHs) after induction of TNF-α significantly reduced cytotoxicity and increased cell survival rates. These findings provide valuable insights into the promising potential of MFHs as a natural source of bioactive peptides relevant to metabolic disorders. To fully realize their application in functional foods, it is essential to identify the specific peptides released after gastrointestinal digestion that contribute to the observed DPP-IV inhibitory activity. Moreover, this study warrants further investigation in in vivo models and clinical settings to validate their physiological effects and therapeutic relevance.

## Figures and Tables

**Figure 1 foods-14-03456-f001:**
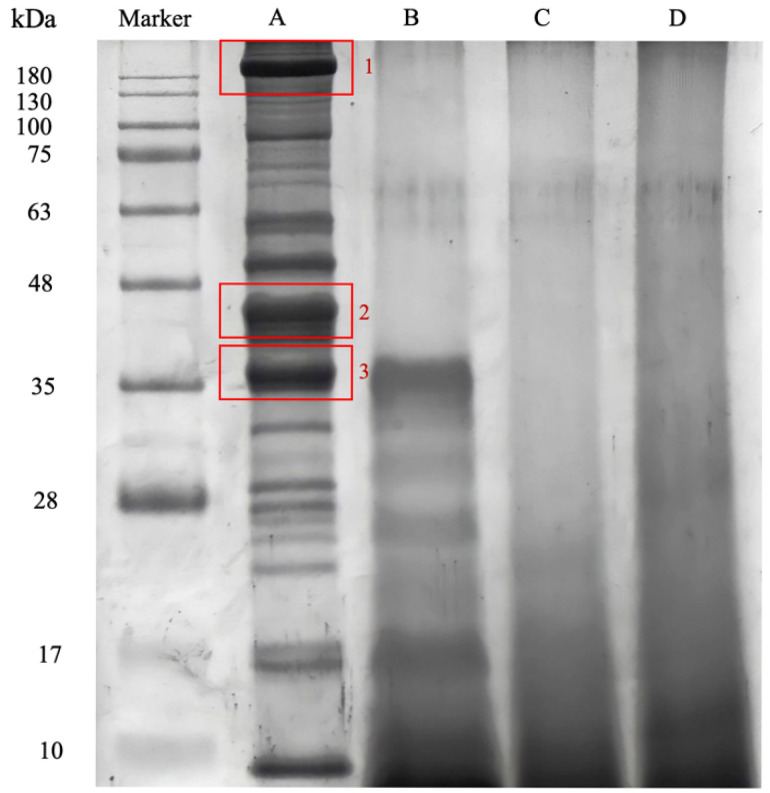
SDS-PAGE of MF protein isolate and hydrolysates. Lane A: MF protein extract; Lane B: pepsin hydrolysate; Lane C: papain hydrolysate; Lane D: bromelain hydrolysate. Molecular weight markers (kDa) are indicated on the left. Highlighted in red are protein bands that were selected for in-gel digestion.

**Figure 2 foods-14-03456-f002:**
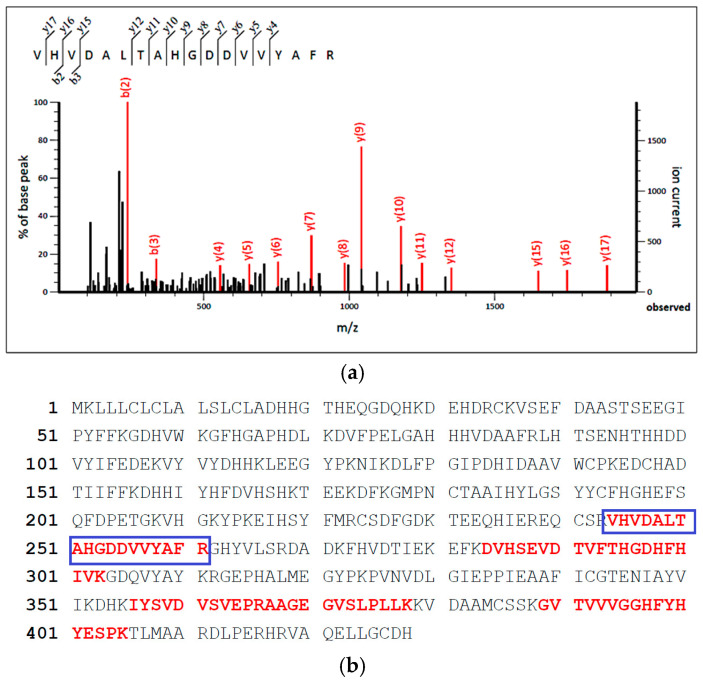
Mass spectrometry identification of bioactive peptides from MFH protein isolates. (**a**) Representative spectra of a triply charged peptide from MF protein isolates (SDS-PAGE bands 1, 2, and 3) (VHVDALTAHGDDVVYAFR, MW: 622.344500,3+) (All annotated peaks exceed the instrument noise threshold, confirming that they represent true signals); (**b**) Sequences of identified tryptic peptides that matched the WAP65-1 (Accession number: ABB08603.1). The blue box highlights the identified peptide sequence.

**Figure 3 foods-14-03456-f003:**
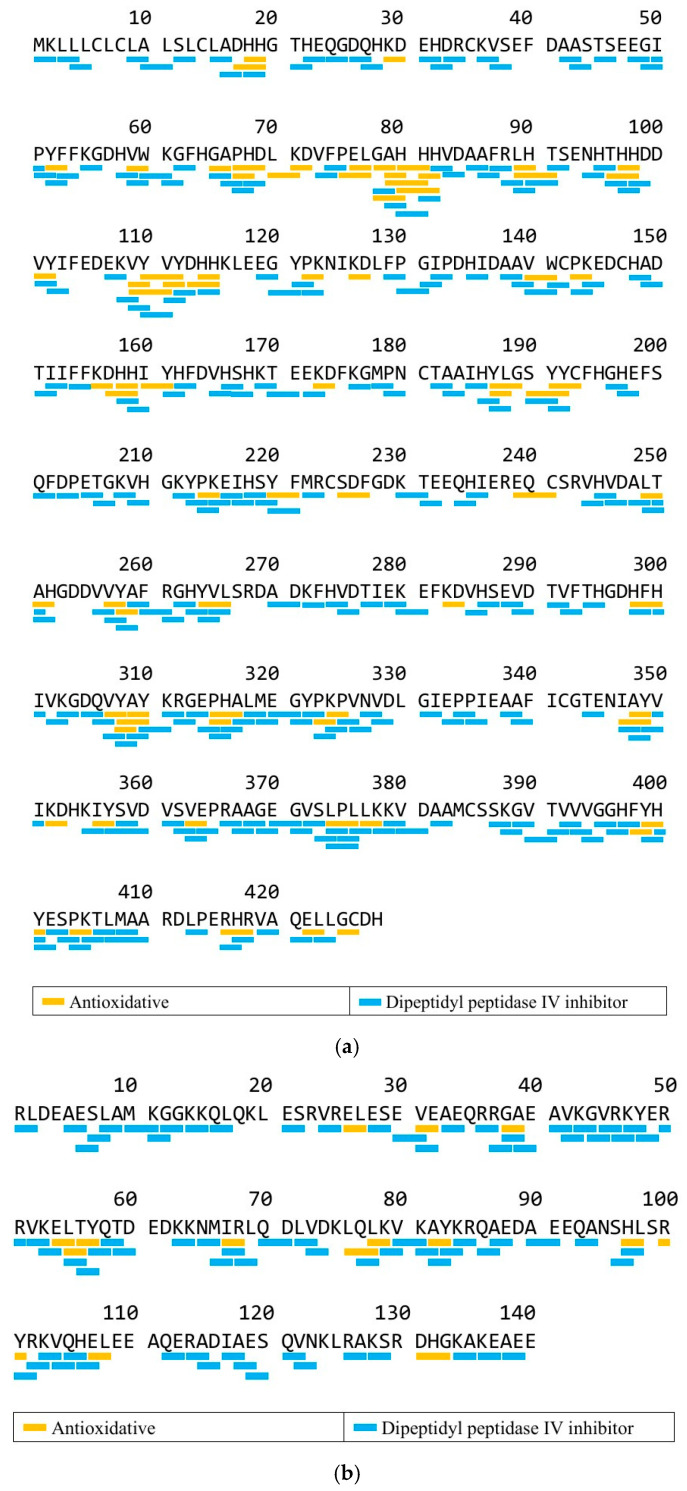
Potential bioactive peptides from MF protein isolate. (**a**) 65 kDa warm temperature acclimation protein 1 (WAP65-1) (Accession number: ABB08603.1); (**b**) myosin heavy chain (Accession number: ABK59967.1).

**Figure 4 foods-14-03456-f004:**
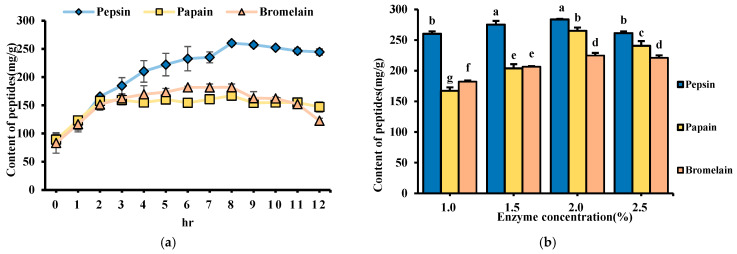
Peptide concentration of hydrolysates produced after enzyme treatment. (**a**) Peptide content at different times, and (**b**) at various concentrations. Different letters indicate statistically significant differences (*p* < 0.05) between groups.

**Figure 5 foods-14-03456-f005:**
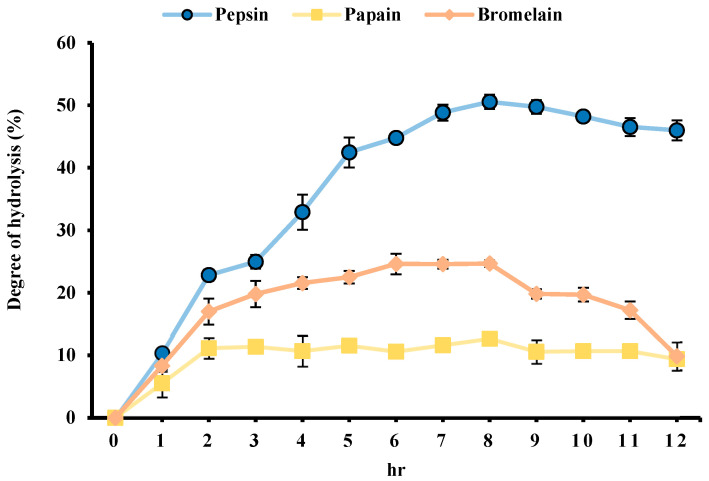
Degree of hydrolysis (DH) of MF powder hydrolyzed using pepsin, papain, and bromelain for 12 h. DH was calculated based on the proportion of peptide bonds cleaved during hydrolysis, reflecting the extent of protein breakdown by each enzyme.

**Figure 6 foods-14-03456-f006:**
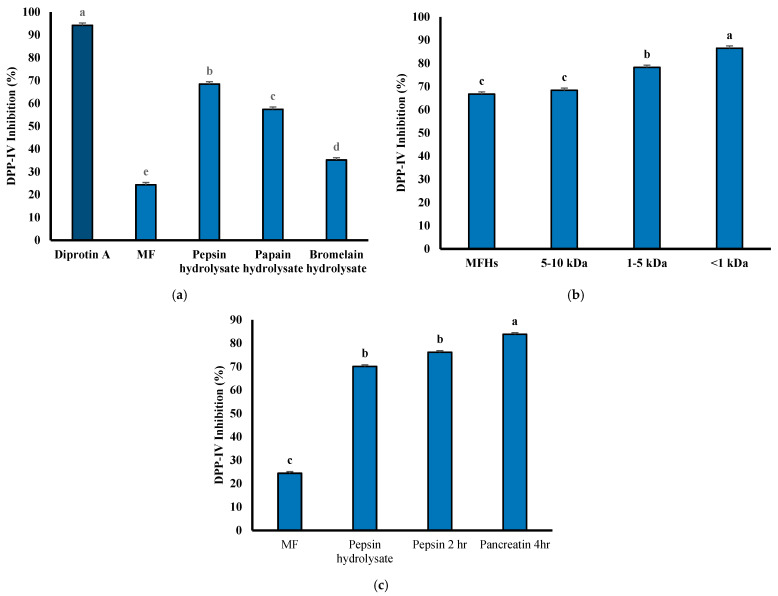
DPP-IV inhibitory activity of MFH. (**a**) DPP-IV inhibitory activity of unhydrolyzed MF and MFH with pepsin, papain, and bromelain; (**b**) pepsin hydrolysates after ultrafiltration; (**c**) pepsin hydrolysates after in vitro simulated gastrointestinal digestion. Inhibitory activity of DPP-IV was evaluated at a final concentration of 10 mg/mL. Values containing different letters are significantly different (*p* < 0.05).

**Figure 7 foods-14-03456-f007:**
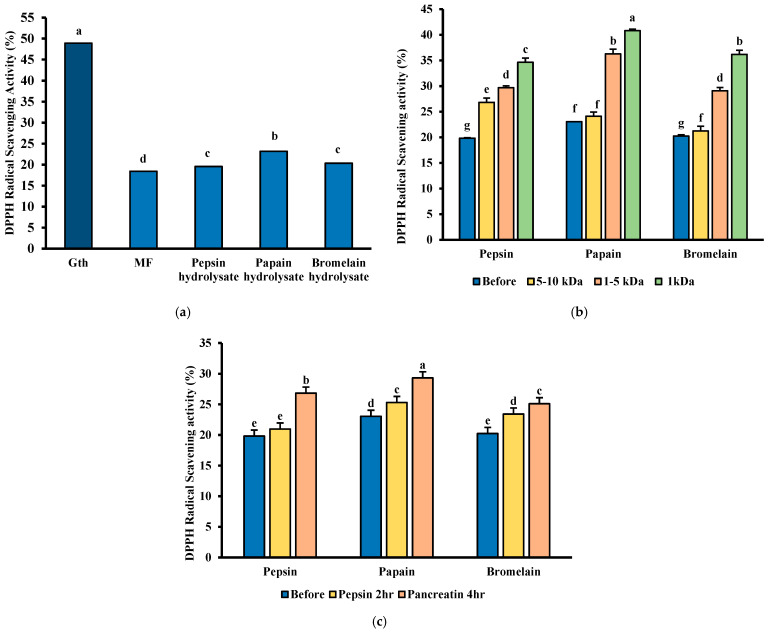
The antioxidant activity of MFH. (**a**) DPPH Radical-Scavenging Activity of MFH; (**b**) DPPH Radical-Scavenging Activity of MFH after ultrafiltration; (**c**) DPPH Radical-Scavenging Activity of MFH after in vitro simulated gastrointestinal digestion. Values containing different letters are significantly different (*p* < 0.05).

**Figure 8 foods-14-03456-f008:**
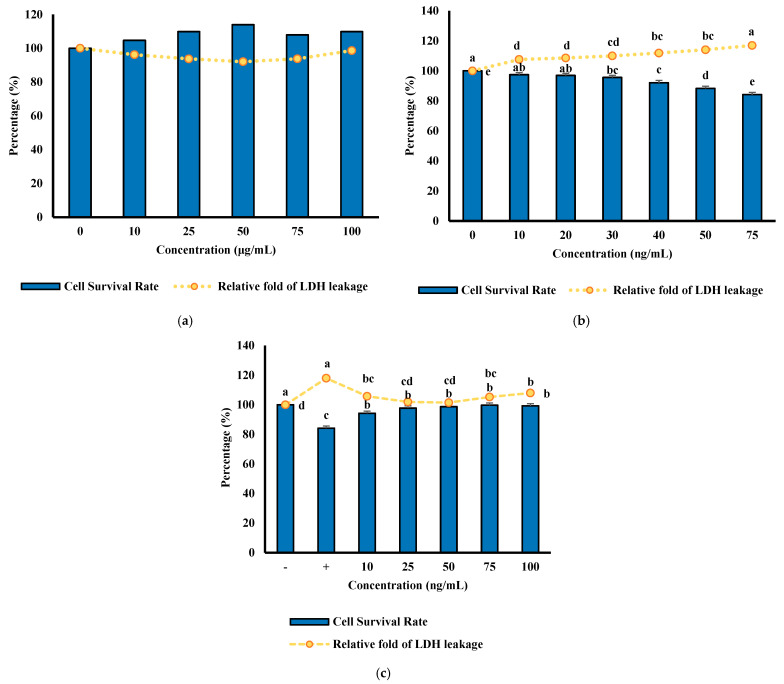
Effects of MFH on FL83B cell viability and protection against TNF-α-induced cytotoxicity. (**a**) Effects of MFHs on cell survival rate and LDH leakage in FL83B cells; (**b**) Effects of TNF-α cell survival rate and LDH leakage in FL83B cells; (**c**) Effects of MFH on TNF-α-Induced FL83B cells and LDH leakage in TNF-α-treated FL83B cells. Values containing different letters are significantly different (*p* < 0.05).

**Table 1 foods-14-03456-t001:** Proximate composition of MF powder.

Composition	Original (%)	After Degreasing (%)
**Moisture**	1.59 ± 0.08	1.76 ± 0.07
**Crude protein**	53.61 ± 0.27	69.15 ± 0.54
**Crude fat**	22.15 ± 0.18	1.08 ± 0.18
**Ash**	11.76 ± 0.31	13.38 ± 0.81
**Carbohydrate**	10.90 ± 0.80	14.63 ± 0.94

Data presented as means ± SD (*n* = 3).

**Table 2 foods-14-03456-t002:** Protein content, peptide content, and yields of MF powder protein isolate.

	Protein Content (%)	Peptide Content (mg/g)	Yield (%) ^1^
**Milkfish frame protein isolate**	71.25 ± 0.71	87.81 ± 3.56	30.05 ± 0.82

Data presented as means ± SD (*n* = 3). ^1^ Yield (%) = (weight after drying/weight before drying) ∗ 100.

**Table 3 foods-14-03456-t003:** Number of predicted bioactive peptides to be generated from the identified MF proteins using the enzyme action tool provided by BIOPEP.

Protease	Myosin Heavy Chain		65 kDA Warm Temperature Acclimation Protein 1 (WAP65-1)	
	DPP-IV Inhibitory	Antioxidative	DPP-IV Inhibitory	Antioxidative
**Ficin**	10	5	30	11
**Papain**	10	1	22	1
**Pepsin (pH > 2)**	15	1	47	16
**Chymotrypsin C**	10	1	16	5
**Proteinase K**	5	1	23	4
**Thermolysin**	6	4	20	1
**Cathepsin G**	3	12	14	5
**Bromelain**	16	3	25	4

**Table 4 foods-14-03456-t004:** Comparison of soluble protein and yields of MF powder with different enzyme concentrations of 1%, 1.5%, 2%, and 2.5% after 8 h of hydrolysis.

Enzyme	Concentration (%)	Soluble Protein (%)	Yield ^1^ (%)
**Pepsin**	1.0	61.80 ± 0.92 ^Ac^	62.69
	1.5	73.79 ± 0.70 ^Ab^	70.44
	2.0	86.46 ± 0.49 ^Aa^	82.54
	2.5	74.00 ± 0.41 ^Ab^	78.42
**Papain**	1.0	39.43 ± 0.52 ^Cc^	32.74
	1.5	56.25 ± 0.47 ^Bb^	40.65
	2.0	59.44 ± 0.39 ^Ca^	63.42
	2.5	27.54 ± 0.58 ^Cd^	61.03
**Bromelain**	1.0	46.85 ± 0.40 ^Bd^	41.43
	1.5	53.23 ± 0.49 ^Cc^	52.56
	2.0	61.28 ± 0.75 ^Ba^	68.25
	2.5	58.66 ± 0.85 ^Bb^	62.11

^1^ Yield (%) = (the weight after drying/the weight before drying) ∗ 100. Data presented as means ± SD (*n* = 3). Values containing different letters (a–c) are significantly different in the same row (*p* < 0.05). Values containing different letters (A–C) are significantly different in the same column (*p* < 0.05).

## Data Availability

The original contributions shown in this study are incorporated within the article. The corresponding author can be directed for further inquiries.

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
