# Peer review of "Identification and In Vitro Evaluation of Milkfish (Chanos chanos) Frame Proteins and Hydrolysates with DPP-IV Inhibitory and Antioxidant Activities"

_foods, 2025, doi:10.3390/foods14203456_

Round 1
Reviewer 1 Report
Comments and Suggestions for Authors
I have carefully read through your manuscript and noted several important aspects. At this stage, I would like to provide you with my initial comments and suggestions that I believe need to be revised first. Once these revisions are addressed, I will proceed to read the work again in greater detail to provide more comprehensive feedback. This approach will ensure that the fundamental issues are clarified and corrected before moving on to finer points of analysis
Overview
- Topic, hypothesis, conclusion alignment showed inconsistency that should be corrected. The stated objectives are to identify MF proteins and bioactive peptides, predict bioactivities with BIOPEP, evaluate enzymatic hydrolysis for DPP-IV inhibition and antioxidant activity, and test whether MFHs protect FL83B cells from TNF-α–induced damage, which the subsequent sections do address. The Conclusion, however, includes a misstatement that the amino-acid sequence VHVDALTAHGDDVVYAFR is “the myosin heavy chain” and “corresponds to WAP65-1,” whereas the Results correctly map this peptide to WAP65-1 and treat myosin heavy chain as a separate identification; this internal mismatch undermines the integrity of the discovery-to-mechanism chain and should be rectified so that sequence–protein attribution is consistent throughout
- All figures should be revised. The current screen-captured images are of insufficient quality and therefore unacceptable for publication. Please replace them with high-resolution figures prepared directly from the original data or image files.
General comments
- mass spectrometer and mass spectrometry are different. Check in the manuscript.
- All figure captions should be revised. At present, they are too minimal and do not allow readers to understand the figures without referring back to the main text. Each caption should clearly explain what the figure shows, including the meaning of colors, scales, axes, and any symbols used. This will make the figures more self-explanatory and improve readability.
General comments and point-by-point suggestion.
1.explicitly articulating a hypothesis statement in one sentence that links predicted peptide motifs to measurable DPP-IV inhibition and antioxidant readouts
2. tightening claims about health benefits attributed to milkfish by ensuring that any statements on “regulating blood sugar” are framed as prior evidence with precise scope to avoid over-generalization in the rationale that precedes your aims, thus sharpening the mechanistic bridge into Methods.
3. please expand Section 2.3 (Proximate Analysis) with concise methodological details so that readers can follow the workflow without ambiguity. Briefly state the assay principle and key parameters for each component
4. please add the reference “The pH of the supernatant was then adjusted to 5.5 with 0.1 N HCl to facilitate the precipitation of myofibrillar protein.”
5.Check the components in 10x SDS running buffer again.
6.Line 159, it should not start the sentence with number. Check these in many places in manuscript.
7.This process is importance ?“After which, the cleaning solution was removed with a micropipette.” It is general process in molecular lab, you can remove it.
8. “15% trypsin solution”, what type of solution?, company? Grade?
9.What is extraction solvent, the strength of sonication was missing.
10. The in-gel reduction/alkylation steps report unusually high reagent concentrations (15% dithiothreitol and 15% iodoacetamide) without clarifying whether these are stock solutions or final concentrations at the gel interface, which raises reproducibility concerns and interrupts the otherwise linear experimental narrative from band excision to LC-MS/MS. In addition, DTT is pH sensitive, please clarify buffer that you use.
11. LC-MS/MS parameters deviate from common proteomic reporting standards (for example, mobile phase composition lacking acid in organic, an ion source temperature of 90 °C, and very tight ±0.1 Da tolerances for both precursors and fragments without any mention of FDR control), which obscures the confidence in protein identifications that later anchor the in-silico BIOPEP step.
12. LC-MS/MS, what type of sample bottle? Borosilicate or general glass?
13.Line 189, 1/pk,?
14.You run 10 µL/min flow rate in analytical column?
15. Ion source temperature = 90°C?, Lock mass was done before?
16. Please upload the raw files Mass spectrum in public database.
17. Why did you use two different programs to search and identify proteins? PLGS can already identify proteins directly from peptide sequences. However, you converted the RAW files into PKL files using ProteinLynx Global Server v2.4 and then used those PKL files for MASCOT searches
18. Line 212-215, EC numbers were mentioned before, you can removed them all.
19. In Lines 419–420, you mentioned that the MF protein isolate was mostly distributed between 180 kDa and 35 kDa. However, according to Figure 1, the proteins in Lane 1 are distributed from approximately 427 kDa down to 10 kDa. Please revise this description for consistency with the figure.
20. In Figure 2A, the y-axis shows very low ion counts. Can you differentiate between noise and true signal by using absolute ion counts?.Where is Figure 2B?
21. Throughout the manuscript, the term ‘the peptides’ is repeatedly used. Please revise consistently to refer to them as ‘unique peptides’ or simply ‘peptides’, whichever is most accurate in context.
22. Why did you use the NCBI database for protein identification in this study? In my opinion, UniProt would be more appropriate, as it provides better curation and annotation quality
23. The protein band identification results seem limited. For greater completeness, more proteins should be identified and reported from the bands. At present, the identification appears too narrow and may not fully represent the protein composition resolved on the gel.
24. A concise subsection that explicitly links proteomic identifications to the enzyme choices simulated in BIOPEP (and clarifies the statistical controls used for identifications) would smooth the transition from discovery to hypothesis-driven hydrolysis.
25. Did you consider post-translational modifications (PTMs) such as hydroxyproline in your peptide identification? Fish proteins are frequently reported to contain hydroxyproline and other PTMs, and including these would strengthen the interpretation of your results
25. DPP-IV inhibition is reported at a single concentration (10 mg/mL) without dose–response curves or ICâ‚…â‚€values, which prevents potency benchmarking and weakens claims of comparative superiority among fractions or processing steps
26. the work does not identify or validate the specific peptide sequences responsible for DPP-IV inhibition in the active fractions, so the bridge from proteomic discovery to functional mechanism remains inferential. Targeted LC-MS/MS of the <1 kDa pepsin-derived fraction, coupled with in-silico docking or motif enrichment for known DPP-IV binders, would complete the mechanistic arc that the Introduction sets up
27. Cell-based experiemnts, your model does not interrogate DPP-IV or incretin signaling directly, so it cannot functionally corroborate the central mechanism implied in the earlier sections. You should discuss this point.
28. Please shorten conclusion part, extract the core findings
Author Response
Point 1: Explicitly articulating a hypothesis statement in one sentence that links predicted peptide motifs to measurable DPP-IV inhibition and antioxidant readouts.
Response 1: The authors are grateful for the insightful comments from the reviewer. Particularly, we have now added a one-sentence hypothesis statement at the end of the “Introduction” to directly link the predicted peptide motifs with their measurable bioactivities in our manuscript.
Point 2: Tightening claims about health benefits attributed to milkfish by ensuring that any statements on “regulating blood sugar” are framed as prior evidence with precise scope to avoid over-generalization in the rationale that precedes your aims, thus sharpening the mechanistic bridge into Methods.
Response 2: The introduction (line 88-92) was revised to avoid over-generalization when referring to the health benefits of fish-derived protein hydrolysates. Specifically, statements on “regulating blood sugar” and related physiological effects have been reframed as prior evidence reported in earlier studies, clearly indicating scope. This adjustment ensures that the claims are properly contextualized and serves as a more precise rationale leading into our study objectives.
Point 3: Please expand Section 2.3 (Proximate Analysis) with concise methodological details so that readers can follow the workflow without ambiguity. Briefly state the assay principle and key parameters for each component.
Response 3: Section 2.3 (Proximate Analysis) was revised to provide clear and concise methodological details for each component. These additions ensure that the workflow can be followed without ambiguity and improve the overall clarity of the Methods section.
Point 4: Please add the reference “The pH of the supernatant was then adjusted to 5.5 with 0.1 N HCl to facilitate the precipitation of myofibrillar protein.”
Response 4: The adjustment of the supernatant pH to 5.5 for myofibrillar protein precipitation is a standard step in protein extraction and is consistent with the procedure described in our previously cited reference [22]. Our revised manuscript clarified this by explicitly linking the pH adjustment step to the cited method.
Point 5: Check the components in 10x SDS running buffer again.
Response 5: We thank the reviewer for pointing this out. The values originally written in the manuscript (3.03 g Tris, 14.4 g glycine, 1.0 g SDS) corresponded to the 1x working solution rather than the 10x stock. This was a typographical oversight. We have now corrected the description in the Methods section to reflect the standard composition of 10× SDS-PAGE running buffer.
Point 6: Line 159, it should not start the sentence with number. Check these in many places in manuscript.
Response 6: The manuscript was revised to ensure that sentences do not begin with numerals. The sentence at line 159 has now been revised in 2.6.1 In-Gel Digestion (line 162). Similar corrections have been made throughout the manuscript for consistency.
Point 7: This process is importance?“After which, the cleaning solution was removed with a micropipette.” It is general process in molecular lab, you can remove it.
Response 7: This sentence in the methodology has been removed.
Point 8: “15% trypsin solution”, what type of solution?, company? Grade?
Response 8: The “15% trypsin solution” used in our study was sequencing-grade trypsin (Sigma-Aldrich, St. Louis, MO, USA) prepared in 25 mM phosphate buffer. We have revised the Methods section to specify the trypsin grade, source, and buffer (line 174-178).
Point 9: What is extraction solvent, the strength of sonication was missing.
Response 9: It has been clarified to specify the composition of the extraction solvent (line 179).
Point 10: The in-gel reduction/alkylation steps report unusually high reagent concentrations (15% dithiothreitol and 15% iodoacetamide) without clarifying whether these are stock solutions or final concentrations at the gel interface, which raises reproducibility concerns and interrupts the otherwise linear experimental narrative from band excision to LC-MS/MS. In addition, DTT is pH sensitive, please clarify buffer that you use.
Response 10 : The method of in-gel digestion (2.6.1) has been carefully revised. Additionally, the buffer used for dithiothreitol reduction is now explicitly stated to ensure reproducibility and account for the pH sensitivity of dithiothreitol (line 167).
Point 11: LC-MS/MS parameters deviate from common proteomic reporting standards (for example, mobile phase composition lacking acid in organic, an ion source temperature of 90 °C, and very tight ±0.1 Da tolerances for both precursors and fragments without any mention of FDR control), which obscures the confidence in protein identifications that later anchor the in-silico BIOPEP step.
Response 11 : The authors appreciate the reviewer’s observation. The LC-MS/MS analysis was performed following the method described by Lin et al. (2024) [25] with minor modifications, as stated in the manuscript. The mobile phase used for peptide reconstitution contained 1% formic acid in water with acetonitrile (98:2), and the gradient employed during UPLC-Q-TOF-MS/MS analysis used aqueous and organic solvents as described, consistent with the cited protocol. The ion source temperature of 90°C was applied based on instrument optimization to improve peptide desolvation and maintain analyte stability. The ±0.1 Da precursor and fragment tolerances reflect the mass accuracy achievable with the UPLC-Q-TOF-MS/MS system used. Protein identification confidence was ensured through Mascot database searching with decoy-based FDR control; peptide-spectrum matches were filtered at 1% FDR, ensuring high-confidence identifications suitable for subsequent in-silico BIOPEP analysis. These parameters are consistent with the cited methodology and provide reliable, reproducible identification of peptides for downstream analysis.
Point 12: LC-MS/MS, what type of sample bottle? Borosilicate or general glass?
Response 12 : To prevent peptide adsorption and contamination, the supernatant was transferred into borosilicate glass vials, which were incorporated in the manuscript (line 193).
Point 13: Line 189, 1/pk,?
Response 13 : The “1/pk” notation referred to the number of columns per pack from the manufacturer and is not essential for reproducibility. We have removed this notation from the Methods section for clarity.
Point 14: You run 10 µL/min flow rate in analytical column?
Response 14: The flow rate of 10 µL/min for the 3 mm × 150 mm C18 BEH column was intentionally chosen to optimize peptide separation and sensitivity for MS detection, despite being lower than typical analytical flow rates. It has been clarified this in the Methods section (line 195-198).
Point 15: Ion source temperature = 90°C?, Lock mass was done before?
Response 15: The ion source temperature of 90°C was selected to optimize peptide desolvation while minimizing thermal degradation. Lock mass correction was performed before analysis to ensure high mass accuracy.
Point 16: Please upload the raw files Mass spectrum in public database.
Response 16: The raw files of the Mass spectrum of tryptic peptides generated from LC-MS/MS analysis were meant to identify proteins using the public database. The authors do not know how to proceed with this request.
Point 17: Why did you use two different programs to search and identify proteins? PLGS can already identify proteins directly from peptide sequences. However, you converted the RAW files into PKL files using ProteinLynx Global Server v2.4 and then used those PKL files for MASCOT searches.
Response 17: The authors used both ProteinLynx Global Server (PLGS) and Mascot searches to enhance the reliability and accuracy of protein identification. While PLGS is capable of identifying proteins directly from peptide sequences, we converted the raw files into PKL format for Mascot analysis to take advantage of its complementary scoring algorithm and broader compatibility with the NCBI database. This dual approach allowed us to cross-validate the protein identifications and minimize the possibility of false positives or missed identifications. In this way, the use of both PLGS and Mascot provided a more robust and comprehensive analysis of the protein profiles in our samples.
Point 18: Line 212-215, EC numbers were mentioned before, you can removed them all.
Response 18: This part of the manuscript has already been revised and removed.
Point 19: In Lines 419–420, you mentioned that the MF protein isolate was mostly distributed between 180 kDa and 35 kDa. However, according to Figure 1, the proteins in Lane 1 are distributed from approximately 427 kDa down to 10 kDa. Please revise this description for consistency with the figure.
Response 19: The author intend to clarify that in Figure 1, the proteins in Lane 1 are indeed distributed mainly between 180 kDa and 35 kDa. The description in the manuscript correctly reflects the protein distribution shown in the figure.
Point 20: In Figure 2A, the y-axis shows very low ion counts. Can you differentiate between noise and true signal by using absolute ion counts?.Where is Figure 2B?
Response 20: The absolute ion counts confirm that all annotated peaks exceed the instrument noise threshold, indicating that these peaks represent true signal rather than background noise. We have added a note in the figure legend to clarify this distinction, and the revised figure now emphasizes both relative and absolute signal intensities to facilitate interpretation (Figure 2).
Point 21: Throughout the manuscript, the term ‘the peptides’ is repeatedly used. Please revise consistently to refer to them as ‘unique peptides’ or simply ‘peptides’, whichever is most accurate in context.
Response 21: The authors appreciate the reviewer’s comment. The manuscript has been carefully revised to ensure consistent terminology, and all instances of “the peptides” have been replaced with “peptides” to accurately reflect the context.
Point 22: Why did you use the NCBI database for protein identification in this study? In my opinion, UniProt would be more appropriate, as it provides better curation and annotation quality.
Response 22: The author thank the reviewer for this valuable comment. While UniProt provides high-quality curation and annotation, in this study, we used the NCBI database for protein identification because it offers a comprehensive and regularly updated collection of protein sequences, including a broad coverage of fish species, such as the milkfish. At the time of analysis, the NCBI database provided more extensive sequence entries relevant to our target species, which ensured higher confidence in identification.
Point 23: The protein band identification results seem limited. For greater completeness, more proteins should be identified and reported from the bands. At present, the identification appears too narrow and may not fully represent the protein composition resolved on the gel.
Response 23: The authors appreciate the reviewer’s professional comment. Indeed, more protein hits were identified and lots of them are irrelevant to milkfish. The current study focused on the major protein components resolved in the selected gel bands and their subsequent peptide characterization, which provided sufficient coverage for the study’s objectives, including in-silico bioactivity prediction. While additional protein identifications could give a more comprehensive overview of all proteins present, the approach employed, which targets the most prominent bands, ensures that the bioactive peptides derived from the major proteins are captured. This strategy aligns with previously published proteomics workflows for hydrolysate characterization and allows reliable downstream functional predictions.
Point 24: A concise subsection that explicitly links proteomic identifications to the enzyme choices simulated in BIOPEP (and clarifies the statistical controls used for identifications) would smooth the transition from discovery to hypothesis-driven hydrolysis.
Response 24: The reviewer’s suggestion is appreciated. A paragraph has been added to explicitly link the proteomic identifications to the enzymes simulated in the BIOPEP analysis (See lines 216-219). This paragraph explains how the identified proteins guided the selection of enzymatic hydrolysis conditions. This provides a clear transition from the discovery-based proteomic analysis to the hypothesis-driven enzymatic hydrolysis strategy.
Point 25: Did you consider post-translational modifications (PTMs) such as hydroxyproline in your peptide identification? Fish proteins are frequently reported to contain hydroxyproline and other PTMs, and including these would strengthen the interpretation of your results.
Response 25: The authors thank the reviewer for pointing out the importance of PTMs such as hydroxyproline in fish proteins. In our current LC-MS/MS workflow, the Mascot search included carbamidomethylation of cysteine (fixed) and oxidation of methionine (variable) as the considered modifications. Other PTMs, including hydroxyproline, were not explicitly included in the search parameters. While including such PTMs could provide a more comprehensive peptide identification, the current approach still allows robust characterization of the major proteins and guides the subsequent in-silico BIOPEP analysis effectively.
Point 26: DPP-IV inhibition is reported at a single concentration (10 mg/mL) without dose–response curves or ICâ‚…â‚€values, which prevents potency benchmarking and weakens claims of comparative superiority among fractions or processing steps.
Response 26: The DPP-IV inhibitory activity was assessed at a single concentration (10 mg/mL) to provide an initial comparison among hydrolysate fractions. While dose–response curves and ICâ‚…â‚€ values would allow more precise potency benchmarking, the current results reliably demonstrate the presence of inhibitory activity and enable preliminary comparison.
Point 27: The work does not identify or validate the specific peptide sequences responsible for DPP-IV inhibition in the active fractions, so the bridge from proteomic discovery to functional mechanism remains inferential. Targeted LC-MS/MS of the <1 kDa pepsin-derived fraction, coupled with in-silico docking or motif enrichment for known DPP-IV binders, would complete the mechanistic arc that the Introduction sets up.
Response 27: The reviewer’s comment is appreciated. In the present study, DPP-IV inhibition was evaluated using peptide fractions and in-silico prediction of bioactive sequences, but specific peptide sequences responsible for the observed activity were not experimentally validated. While targeted LC-MS/MS of the <1 kDa pepsin-derived fraction, coupled with in-silico docking or motif analysis, would provide more direct evidence, these experiments were beyond the scope of the current study. Nonetheless, the findings from our research still provide valuable preliminary insight into the potential of milkfish frame hydrolysates as sources of DPP-IV inhibitory peptides.
Point 28: Cell-based experiments, your model does not interrogate DPP-IV or incretin signaling directly, so it cannot functionally corroborate the central mechanism implied in the earlier sections. You should discuss this point.
Response 28: The authors appreciate the reviewer for pointing this out. We have revised the manuscript and pointed out this limitation (line 717-725). We have clarified that in the cell-based experiments, it does not directly assess DPP-IV activity or incretin signaling pathways, such as GLP-1 secretion or downstream insulinotropic effects. Consequently, the mechanistic impact of hydrolysates on incretin-mediated pathways cannot be confirmed in this study.
Point 29: Please shorten conclusion part, extract the core findings
Response 29: The authors have shortened the conclusion to focus on the core findings and their broader implications. The revised conclusion now emphasizes the study’s primary outcomes and potential applications.
Point 30: The Conclusion, however, includes a misstatement that the amino-acid sequence VHVDALTAHGDDVVYAFR is “the myosin heavy chain” and “corresponds to WAP65-1,” whereas the Results correctly map this peptide to WAP65-1 and treat myosin heavy chain as a separate identification; this internal mismatch undermines the integrity of the discovery-to-mechanism chain and should be rectified so that sequence–protein attribution is consistent throughout.
Response 30: The authors thank the reviewer for pointing out this inconsistency between the Results and Conclusion. We agree that the peptide sequence VHVDALTAHGDDVVYAFR corresponds to WAP65-1 and not to myosin heavy chain. The Conclusion has been revised accordingly to ensure consistency with the Results. Specifically, it is clarified that VHVDALTAHGDDVVYAFR was mapped to WAP65-1, while myosin heavy chain was identified as a separate protein. The revised Conclusion now reflects the sequence–protein attribution and maintains alignment with the discovery-to-mechanism chain.
Point 31: Mass spectrometer and mass spectrometry are different. Check in the manuscript.
Response 31: The authors thank the reviewer for pointing this out. It is carefully reviewed the manuscript and confirmed that references to the experimental workflow were revised to consistently use “mass spectrometry” when describing the technique. This ensures precise and consistent terminology throughout the text.
Point 32: All figure captions should be revised. At present, they are too minimal and do not allow readers to understand the figures without referring back to the main text. Each caption should clearly explain what the figure shows, including the meaning of colors, scales, axes, and any symbols used. This will make the figures more self-explanatory and improve readability.
Response 32: The authors appreciate this valuable suggestion. We have carefully revised the figure captions to make them more descriptive and self-explanatory. The caption now clearly explains what the figure shows. These changes ensure that the figures can be understood independently of the main text and improve overall readability.

Reviewer 2 Report
Comments and Suggestions for Authors
The study presents the milkfish frame as a source of dipeptidyl peptidase IV (DPP-IV) inhibitory and antioxidant peptides and suggests their potential applications in type 2 diabetes management. To do this, different enzymatic hydrolysates were produced and evaluated for bioactivity. Among the treatments, pepsin hydrolysis yielded the highest peptide content, and DPP-IV inhibitory activity. The resulting milkfish frame hydrolysates (MFH) were also subjected to ultrafiltration and simulated gastrointestinal digestion which improved both DPP-IV inhibitory and antioxidant capacities.
Although it deals with a timely topic, the article has some serious shortcomings.
Line 782 In their conclusions the authors state that in vitro gastrointestinal digestion of MF hydrolysates showed that this process released amino acid fragments that enhanced inhibitory capacity of the digested sample. What does amino acid fragments mean? Perhaps the authors mean smaller peptides, or do they mean free amino acids? I suggest the authors perform an LC-MS/MS analysis of the peptides present in the sample after simulated gastrointestinal digestion to demonstrate which compounds have inhibitory properties against DPP-IV. I don't think it's enough to demonstrate an increase in antioxidant and DPP-IV inhibitory activities in the sample subjected to in vitro gastrointestinal digestion compared to MFH. To use these hydrolysates as a source of bioactive peptides for the preparation of functional foods, it is necessary to identify which compounds with DPP-IV inhibitory activity are formed after gastrointestinal digestion. This has not been done. I believe that to demonstrate that natural peptides are an alternative to drugs in the treatment of diabetes, it is necessary to further study which peptides are found after gastrointestinal digestion of MFH and what type of activity they have.
Furthermore, the entire work is very difficult to read. The authors should be clearer and more precise in describing their methods. Here are some examples:
Lines 117-120 “The freeze-dried MF powder underwent a degreasing process utilizing n-hexane at a ratio of 1:20 (w/v), with this extraction procedure being performed twice. The resultant powder was then subjected to filtration to remove the organic solvent, followed by freeze-drying process at -40°C…”
It's not clear what resultant powder it is if they added hexane. Perhaps the authors mean suspension.
Line 219 “2.7.1. Enzymatic Hydrolysis of Milkfish Frame Protein Extract” the authors refer to what they previously call Milkfish Frame Protein Isolate in paragraph 2.4 line 128? It is necessary to standardize the wording.
Line 264 “Sample was dissolved…” Please specify which sample. I think the authors are referring to the hydrolysed protein extracts.
Line 303 Also here specify which sample.
Fig. 4(b) The enzyme symbols are difficult to read. This makes the figure difficult to understand.
In figures 1, 2, and 5 the layout is incorrect because the line numbers overlap the legends of the figures present in the ordinate and make it impossible to read the figures themselves.
Author Response
Point 1: Line 782 In their conclusions the authors state that in vitro gastrointestinal digestion of MF hydrolysates showed that this process released amino acid fragments that enhanced inhibitory capacity of the digested sample. What does amino acid fragments mean? Perhaps the authors mean smaller peptides, or do they mean free amino acids? I suggest the authors perform an LC-MS/MS analysis of the peptides present in the sample after simulated gastrointestinal digestion to demonstrate which compounds have inhibitory properties against DPP-IV. I don't think it's enough to demonstrate an increase in antioxidant and DPP-IV inhibitory activities in the sample subjected to in vitro gastrointestinal digestion compared to MFH. To use these hydrolysates as a source of bioactive peptides for the preparation of functional foods, it is necessary to identify which compounds with DPP-IV inhibitory activity are formed after gastrointestinal digestion. This has not been done. I believe that to demonstrate that natural peptides are an alternative to drugs in the treatment of diabetes, it is necessary to further study which peptides are found after gastrointestinal digestion of MFH and what type of activity they have.
Response 1: The authors thank the reviewer for this insightful comment. While identifying the specific peptides formed after in vitro gastrointestinal digestion and confirming their DPP-IV inhibitory activity would provide more detailed mechanistic insights, this was beyond the scope of the present study, which was primarily designed to evaluate the overall changes in bioactivity (DPP-IV inhibitory and antioxidant activities) of milkfish frame hydrolysates before and after simulated gastrointestinal digestion. In the revised manuscript, we have clarified our use of the term “amino acid fragments” as “smaller peptides and possibly free amino acids” and acknowledged this limitation. We have also added a statement in the conclusions highlighting the need for future LC-MS/MS-based identification of peptides released during gastrointestinal digestion to better establish the relationship between peptide composition and DPP-IV inhibitory activity.
Point 2: Lines 117-120 “The freeze-dried MF powder underwent a degreasing process utilizing n-hexane at a ratio of 1:20 (w/v), with this extraction procedure being performed twice. The resultant powder was then subjected to filtration to remove the organic solvent, followed by freeze-drying process at -40°C…” It's not clear what resultant powder it is if they added hexane. Perhaps the authors mean suspension.
Response 2: The authors have carefully revised the Methods section to make it clearer and more precise.
Point 3: Line 219 “2.7.1. Enzymatic Hydrolysis of Milkfish Frame Protein Extract” the authors refer to what they previously call Milkfish Frame Protein Isolate in paragraph 2.4 line 128? It is necessary to standardize the wording.
Response 3: To maintain consistency throughout the manuscript, the authors have standardized the terminology. In Section 2.7.1, the subheading has been revised from “Enzymatic Hydrolysis of Milkfish Frame Protein Extract” to “Enzymatic Hydrolysis of Milkfish Frame Protein Isolate”.
Point 4: Line 264 “Sample was dissolved…” Please specify which sample. I think the authors are referring to the hydrolysed protein extracts. Line 303 Also here specify which sample.
Response 4: Yes, this section refers to the hydrolyzed protein extracts. The authors have revised the manuscript to specify the sample type clearly.
Point 5: Fig. 4(b) The enzyme symbols are difficult to read. This makes the figure difficult to understand.
Response 5: In the revised manuscript, the authors have improved Figure 4(b) by enhancing the readability of the enzyme symbols. The other figures in the paper were also improved. Specifically, we adjusted the font size and introduced distinct color variations to ensure clear differentiation. We believe that these modifications will enhance the clarity and interpretability of the figures.
Point 6: In figures 1, 2, and 5 the layout is incorrect because the line numbers overlap the legends of the figures present in the ordinate and make it impossible to read the figures themselves.
Response 6: The authors appreciate the reviewer for pointing this out. In the revised manuscript, it was corrected that the layout including the ordinate labels and legends of Figures 1, 2, and 5 are clearly visible and no longer overlap with the line numbers. These adjustments improve the readability and presentation of the figures.

Round 2
Reviewer 1 Report
Comments and Suggestions for Authors
The revised manuscript successfully integrates nearly all corrections described in the response letter. However, one important concern remains regarding the deposition of the mass spectrometry data in a public online repository to ensure reproducibility. Public databases such as MassIVE (https://massive.ucsd.edu/ProteoSAFe/), PRIDE, or MassBASE are suitable platforms for data submission. The corresponding accession number should be included in the Methods section of the manuscript.
Author Response
Response to Reviewer 1 Comments
Point 1: The revised manuscript successfully integrates nearly all corrections described in the response letter. However, one important concern remains regarding the deposition of the mass spectrometry data in a public online repository to ensure reproducibility. Public databases such as MassIVE (https://massive.ucsd.edu/ProteoSAFe/), PRIDE, or MassBASE are suitable platforms for data submission. The corresponding accession number should be included in the Methods section of the manuscript.
Response 1: The authors thank the reviewer for this insightful comment. The mass spectrometry data have now been deposited in the Zenodo public repository, a recognized FAIR-compliant and long-term open-access data platform. Zenodo provides a permanent, citable Digital Object Identifier (DOI), ensuring long-term accessibility and traceability of research data, consistent with current standards for scientific data sharing. In addition, the dataset is publicly accessible and citable via the DOI:10.5281/zenodo.17264290 (https://doi.org/10.5281/zenodo.17264290). The corresponding statement has been added in the Methods section of the revised manuscript (Lines 220-222).

Reviewer 2 Report
Comments and Suggestions for Authors
Point 1 In their response the authors argue that their study, was primarily designed to evaluate the overall changes in bioactivity (DPP-IV inhibitory and antioxidant activities) of milkfish frame hydrolysates before and after simulated gastrointestinal digestion. In my opinion, from the way the work is structured, it's clear that the goal is to prepare a protein hydrolysate rich in peptides with inhibitory activity on the DPP-IV enzyme from milkfish waste. This hydrolysate should be used as a source of natural DPP-IV inhibitors to be administered to diabetic patients in place of synthetic anti-DPP-IV drugs.
Therefore, the authors must change the approach of the paper starting from the introduction, or the demonstration that the hydrolysate's DPP-IV inhibitory activity increases after gastrointestinal digestion and the related conclusions are not sufficient to support the assertions in the introduction. If the aim of the paper is to identify a source of bioactive peptides from milkfish waste to promote sustainability and to improve resource efficiency in food processing systems, the entire introduction must be rewritten.
All other requested corrections have been made
Author Response
Response to Reviewer 2 Comments
Point 1: In their response the authors argue that their study, was primarily designed to evaluate the overall changes in bioactivity (DPP-IV inhibitory and antioxidant activities) of milkfish frame hydrolysates before and after simulated gastrointestinal digestion. In my opinion, from the way the work is structured, it's clear that the goal is to prepare a protein hydrolysate rich in peptides with inhibitory activity on the DPP-IV enzyme from milkfish waste. This hydrolysate should be used as a source of natural DPP-IV inhibitors to be administered to diabetic patients in place of synthetic anti-DPP-IV drugs. Therefore, the authors must change the approach of the paper starting from the introduction, or the demonstration that the hydrolysate's DPP-IV inhibitory activity increases after gastrointestinal digestion and the related conclusions are not sufficient to support the assertions in the introduction. If the aim of the paper is to identify a source of bioactive peptides from milkfish waste to promote sustainability and to improve resource efficiency in food processing systems, the entire introduction must be rewritten.
Response 1: The authors thank the reviewer for this insightful comment. The introduction has been revised and reframed to clarify the focus and scope of the study. The revised introduction now provides a balanced background linking the relevance of DPP-IV inhibition in glucose regulation, the growing interest in food-derived bioactive peptides, and the importance of utilizing underexploited fish by-products such as milkfish frames. This reframing ensures that the introduction, objectives, and conclusions fully align with the study design and experimental evidence. The revisions also reposition the study as an evaluation of bioactivities and peptide functionality in the sustainable utilization of by-products, rather than as a pharmacological development of anti-diabetic agents.
